# Reconstruction of drought and long-rain chronologies since the 17th century in Central Japan using intra-annual tree-ring oxygen isotope ratios and documentary records

Hiroto Iizuka[1], Kenjiro Sho[1], Zhen Li[2], Masaki Sano[3], Yoshikazu Kato[2], Takeshi Nakatsuka[2]

[1]Department of Architecture, Civil Engineering, and Industrial Management Engineering, Nagoya Institute of Technology, Nagoya 466-8555, Japan
[2]Graduate School of Environmental Studies, Nagoya University, Nagoya 464-8601, Japan
[3]National Museum of Japanese History, Chiba 285-8502, Japan

*Correspondence to*: Kenjiro Sho (show@nitech.ac.jp)

**Abstract:** Oxygen isotope ratios ($\delta^{18}O$) of tree-ring cellulose and historical documentary records are widely used to explore the hydroclimatic conditions of the past. In this study, we attempted to reconstruct chronologies of local climate disasters spanning four centuries in Central Japan using these proxy data. For tree-ring $\delta^{18}O$ measurements, we prepared cellulose samples from a long-living cedar tree with continuously broad ring widths. To enhance the temporal resolution, we divided each annual ring into several (mainly six) segments. Analysis of correlations with observed relative humidity and precipitation data revealed that the intra-ring $\delta^{18}O$ variations of the sample tree reflected the hydroclimatic conditions from April to July in each year. Subsequently, we chronologically listed the occurrence of eight types of disasters in the 17th to 19th centuries in the area adjacent to the sample tree according to 20 titles of "Town/City history," compilation of historical documentary records from the local municipality. By comparison with the intra-ring $\delta^{18}O$ data, we found that most of the major droughts and long rains recorded in the historical documents occurred in the Baiu rainy season (typically June–July) or pre-Baiu season that correspond to the growing season of the sample tree. Based on analysis of intra-ring $\delta^{18}O$ variation for documentary-based drought and long-rain years, we set thresholds of intra-ring $\delta^{18}O$ values to identify and extract drought and long-rain years. Drought and long-rain chronologies obtained by applying these thresholds were temporally continuous, complementing those based on documentary records. They depicted the relationships between the frequency of these climate disasters and the occurrence of major famines and the long-term tendency of length and magnitude of the Baiu rainy season in historical times.

## 1 Introduction

Attempts to reconstruct climatic conditions before the installation of modern meteorological observations have utilized various types of proxy data, such as coral skeletons, lake bottom sediments, tree rings, and historical documents. In particular, tree rings and historical documentary records are used in Japan due to their high temporal resolution and availability. In recent years, the oxygen-stable isotope ratio ($\delta^{18}O$) of tree-ring cellulose has been extensively studied in paleoclimatology, since it is strongly influenced by atmospheric relative humidity and precipitation (Rebetez et al., 2003; Liu et al., 2010; Liu et al., 2019; Pumijumnong et al., 2020). This effect is due to the isotope fractionation that occurs during the transpiration of water in tree leaves (Sternberg et al., 1986; Roden et al., 2000). In Japan, a negative correlation between tree-ring $\delta^{18}O$ and precipitation in the growing season has also been reported (Li et al., 2015; Uemura et al., 2018), which enhances the relationship between tree-ring $\delta^{18}O$ and the local hydroclimatic condition. Though the isotope ratio measurements are mainly performed annually (Kurita et al., 2016; Sakashita et al., 2017; Uemura et al., 2018), some studies have categorized each annual ring into equally spaced earlywood and latewood rings (Xu et al., 2016; Xu et al., 2020; Watanabe et al., 2023). These studies attained a higher temporal resolution than those of earlier studies based on the measurement of whole annual rings.

However, only a few studies practically employed intra-ring $\delta^{18}O$ measurement for paleoclimatic reconstruction due to the time-consuming process of extracting and separating cellulose from tree samples, as well as the limited availability of equipment capable of measuring $\delta^{18}O$ in tree-rings.

Historical documentary records contain various types of information relating to the climate, such as daily weather and annual crop yields (Thomson et al., 2019; Nash et al., 2021). For instance, harvest records of wine grapes were collected and used for the reconstruction of spring and summer climates for centuries in Europe (Chuine et al., 2004; Cook and Wolkovich, 2016). In Japan, weather records in old diaries have been used to reconstruct various climate parameters such as summer temperature (Mikami,1993; Hirano et al., 2013), winter climate patterns (Hirano and Mikami, 2008; Hirano et al., 2022), global solar radiation (Ichino et al., 2018) the duration of the rainy season (Mizukoshi, 1993; Mizukoshi, 2001; Sho et al., 2017), and the magnitude of drought (Sho and Tominaga, 2004). Annually recorded cherry bloom dates were also used to reconstruct early spring temperatures since the Heian era (794–1192) in Kyoto (Aono and Saito, 2010; Aono, 2011) and since the Edo era (1603–1868) in Tokyo (Aono, 2015). Maejima and Tagami (1986) collected records on various types of disasters since the 7th century to analyze summer and winter conditions using the charts of climatic hazard distribution. Many of these earlier studies used original historical documents that were continuously reported over a long period. However, collecting these historical documents that usually sparsely contain useful information for climatic reconstruction and compiling those descriptions into objective form requires an enormous amount of time and effort, making it difficult to obtain a long-term, temporally consistent chronology of climate parameters comparable to other proxy or observed data. Consequently, very few studies have compared annual tree-ring data with historical documentary records (Fukuoka, 1993). This study focuses on disaster records compiled in "Town/City histories," which are published by many municipalities in Japan to summarize the local history. They often contain descriptions of disasters that occurred in each town/city, which can be used efficiently to compose a chronicle of climate disasters such as anomalous dry/wet conditions. However, its continuity and homogeneity are not guaranteed because it is based on discrete records excerpted from historical documents with the possibility of many missing or biases.

In this study, therefore, we attempted to combine tree-ring $\delta^{18}O$ data and Town/City history records to explore the local history of climate disasters for about 400 years in the central part of Japan. We conducted intra-annual tree-ring $\delta^{18}O$ analysis to obtain a high temporal resolution and to complement the discontinuous disaster chronologies based on documentary records.

## 2 Materials and methods

### 2.1 Study site

The tree-ring sample and documentary records used in this study were collected in the Tono area, Japan. Tono area is the common name for the eastern part of the Gifu Prefecture which is located inland on the Pacific side of Central Japan (Fig. 1). It is a mountainous area of over 1,560 km$^2$ and the elevation ranges from 100 to 1,000 m above sea level. The climate is hot and humid in summer and cold in winter. The average annual temperature and annual precipitation are approximately 13 °C and 2,000 mm, respectively. It has a humid subtropical climate with high summer temperatures (Fig. 2). Rainfall also increases during summer. Additionally, this area experiences a rainy season called "Baiu" with intermittent rainfall typically from early June to mid-July. Typhoons that develop off the southern oceans strike mainly in late August and September. Analysis of the average monthly temperature and precipitation between 1883 and 2020 shows that the Baiu season is responsible for the high precipitation in June and July, and typhoons are responsible for the high precipitation in September.

## 2.2 δ¹⁸O analysis of tree-ring sample

The tree sample used in this study is a Japanese cedar (*Cryptomeria japonica*) that grew at Okute-Shinmei Shrine (137°17'38" E, 35°25'58" N, 510 m a.s.l.) in Mizunami City in the Tono area, Gifu Prefecture, Japan (Fig. 1c). This tree was approximately 40 m height, 3.5 m DBH, 670 years old, and fell in July 2020. Disk samples were collected at a height of approximately 20 m (Fig. 3). As the inside of the trunk was heavily corroded, a sample corresponding to approximately 400 years (1609–2020) was used in this study. Japanese cedar is an evergreen coniferous tree that grows in most of Japan except for the northernmost and southernmost areas. The growth period of cedar trees in Gifu Prefecture is estimated to range from May to August (Hirano et al., 2020).

The "plate method" was used to extract α-cellulose from tree-ring samples (Kagawa et al., 2015; Xu et al., 2011). Plate samples can be directly made into α-cellulose-only samples through chemical treatment. A thin plate along the transverse section (thickness and width were 1.5 and 10 mm, respectively) was cut from the disk sample using a diamond wheel saw. Thin-plate samples were bleached using a sodium chlorite solution. The hemicellulose was removed using a sodium hydroxide solution and washed with an organic solvent. Samples were dried and then cut into intra-annual segments using an ophthalmic knife and microscope. Each annual ring was evenly divided into six or more segments from 1609 to 1949, except for several years. The tree-ring widths of this sample were consistently > 1 mm in most years until 1949. Six is the practical limit of dividing a 1-mm ring width with our current technique; therefore, we divided each annual ring into six segments as long as possible. Each ring was divided into 12 segments for 50 years of 1609–1683 period, and into 2 segments for 52 years of 1925–2013 period. Annual measurement was conducted for 7 years (1969–1975) since the ring widths were too narrow (Fig. 4a).

Cellulose samples from each segment were weighed on an electronic balance to approximately 150 μg. We determined ¹⁸O/¹⁶O ratios (R) using isotope ratio mass spectrometry interfaced with a pyrolysis-type elemental analyzer (TCEA/Delta V Advantage; Thermo Fisher Scientific, Massachusetts, USA) at the Graduate School of Environmental Studies, Nagoya University, Nagoya, Japan. The ¹⁸O/¹⁶O ratios were compared to those of a cellulose standard (Merck cellulose).

Stable isotopes were expressed as the deviation of the stable isotope ratio of the analyzed sample from the international reference material in terms of per mil (‰) using the following equation:

$$\delta^{18}O = \left(\frac{R_{sample}}{R_{standard}} - 1\right) \times 1000 \ [‰] \quad (1)$$

where $R_{sample}$ and $R_{standard}$ are the ¹⁸O/¹⁶O ratios of the sample and the standard, the Vienna Standard Mean Ocean Water (VSMOW), respectively. The oxygen isotope ratio data were presented using this delta notation. The mean standard deviation of multiple measurements for the same segment of the same tree ring was ± 0.38‰ (n = 816). This is larger than the typical analytical uncertainties for annual tree-ring samples of ± 0.2‰ (Sano et al., 2023; Zhao et al., 2023) because the value of standard deviation for this study includes the influence of variability in δ¹⁸O within tree-ring in addition to the error in isotope measurement. However, this value is sufficiently smaller than the typical range of intra-ring δ¹⁸O variation of 4–6‰ (Fig. 4c).

## 2.3 Disaster records in Town/City history documents

"Town/City histories" are published in most municipalities in Japan to document and summarize the local history focusing on their lifestyle, climate, economy, and other factors and often contain descriptions of disasters excerpted from historical documents left in the town/city. The descriptions of disasters in these historical documents are subjective, depending on the writer. Most of these descriptions report only disaster occurrences, although some contain other details, such as the season and duration of the occurrence. Furthermore, the continuity of these records is not guaranteed in many cases. A lack of disaster records for a certain year does not necessarily indicate the non-occurrence of significant disasters in that year. Other reasons

may explain the missing disaster record for that year. In this study, we collected as many disaster records as possible from Town/City histories in the vicinity area of the sample tree to evaluate the magnitude of disasters quantitatively.

All 20 titles of Town/City histories in the Tono area were used in this study (Fig. S1 and Table S1). Disaster descriptions were extracted from these sources and categorized according to title and disaster type in chronological tables. The eight types of disaster categorized in this study were long rain, drought, flood, heavy rain, bad harvest, famine, heavy wind, and insect damage. The covered period ranged from 1601 to 1900, for which, all the tree rings could be divided into six or more segments except for several years. Table 1 shows the total number of disaster descriptions for each disaster type for each 50-year interval. The total number of disaster descriptions for each year is shown in Table S2.

## 3.    Results and discussion

### 3.1 Long-term variation in the annual $\delta^{18}$O data

The results of the measurement of intra-ring $\delta^{18}$O of the sample tree are shown in Fig. 4. The 15-year moving average of $\delta^{18}$O in Fig. 4b (black line) showed a decreasing trend from 1650 to 1790, followed by an increasing trend from 1860 to 1920. This is consistent with Sakashita et al. (2017), who reported high oxygen isotope ratios in the 1601–1700 period and low ratios in the 1750–1850 period for a Japanese cedar sampled 120 km south of the sample tree of this study. Nakatsuka et al. (2020) investigated the long-term variation of the climatological component of tree-ring $\delta^{18}$O in Central Japan. Their resultant chronology was compared with the climatological component of the annual averaged tree-ring $\delta^{18}$O of the sample tree of this study calculated as described by Nakatsuka et al. (2020) (Fig. S2). The two chronologies showed similar long-term variations for almost the entire measurement period of this study, along with a significant positive correlation, r = 0.52 (n = 395, p < 0.01). These results indicate that the tree-ring $\delta^{18}$O variations obtained in this study reflect the regional common trend in Central Japan.

The range of fluctuation in $\delta^{18}$O within each annual ring was approximately 6–8‰ before 1660 for which the number of segments per ring is mainly 12 (orange line in Fig. 4a). Meanwhile, the range was 0.7–4‰ in most years after 1940 for which many of annual rings were divided into two segments or not divided at all (measured annually). The range of intra-annual fluctuation in $\delta^{18}$O closely reflected the number of segments per ring.

### 3.2 Correlation between intra-ring $\delta^{18}$O and meteorological observation data

$\delta^{18}$O measurements were correlated with relative humidity and precipitation data observed at the nearest meteorological station from the sampling site (Gifu Local Meteorological Office, 48 km west of the sampling site, Fig. 1c). The period of the growing season of cedar trees in Central Japan is reported as May-August (Hirano et al., 2020). Since monthly meteorological data may be too coarse to analyze the correlations with the $\delta^{18}$O data corresponding to six separate periods of the growing season, we used the 10-day data commonly used as sub-monthly meteorological statistics in Japan.

The correlation coefficients between intra-ring $\delta^{18}$O values and the 10-day relative humidity and precipitation for each segment for the years when six-divided $\delta^{18}$O data are available within the observation period since 1883 are shown in Fig. 5. The negative correlation peaked in mid-April to early May for both relative humidity and precipitation in the first segment (Fig. 5a). The peak of the negative correlation gradually moved to the later season, and the highest correlation was found in mid-July in the fifth segment (Fig. 5e). For the sixth segment, no specific 10-day period was found when the correlation with both relative humidity and precipitation is significant (Fig. 5f). This may be because, at least partly, the earliest part of wood from the following ring was mixed with the sample of the sixth segment due to difficulty in accurate identification of the ring boundary in some rings, which obscures the peak of correlation with relative humidity or precipitation in specific 10-day

period. Consequently, the growing season of the sample tree was estimated roughly as mid-April to mid-July. This is consistent with the growth period of cedar trees in Central Japan identified in previous studies (Kawana et al., 1973; Hirano et al., 2020). This result indicates the intra-ring $\delta^{18}O$ data in this study reflect the seasonal variation of local hydroclimatic conditions from spring to early summer including the Baiu rainy season.

### 3.3 Comparison of intra-ring $\delta^{18}O$ variation and documentary disaster records

We compared the seasonal variation of $\delta^{18}O$ by eight disaster types classified as shown in Table 1. The mean values of $\delta^{18}O$ for each segment for each of the years of long rain, drought, flood, and famine are shown in Fig. S3. Herein, years of each type of disaster were extracted as years for which the disaster record is found in $\geq 3$ titles out of 20 Town/City histories. The reason for using $\geq 3$ titles as the threshold is described in 3.4.3. The period of analysis was set as 1701–1900 as the disaster records are very sparse in the 17th century compared with the 18–19th centuries (Table 1).

The $\delta^{18}O$ values were generally higher in drought years and lower in long-rain and flood years than the average of all years for 1701–1900 (Fig. S3). Especially, the $\delta^{18}O$ average in drought years significantly differed from the $\delta^{18}O$ average during 1701–1900 for the 4th to 6th segments ($p < 0.05$). Furthermore, the $\delta^{18}O$ average in long-rain years significantly differed from the $\delta^{18}O$ average during 1701–1900 for the 1st and 5th segments ($p < 0.05$) and the 4th segment ($p < 0.10$).

This is consistent with the negative correlation between tree-ring $\delta^{18}O$ and relative humidity (precipitation) in the growing season. Intra-ring $\delta^{18}O$ variation for long-rain and flood years were similar (Fig. S3) since both were caused by heavy precipitation, although did not necessarily coincide. None of the nine long-rain and six flood years recorded in $\geq 3$ titles of Town/City history during the 1701–1900 period occurred in the same year (Table S2), probably because long rains and floods are linked to rainfall events in different time scales.

Meanwhile, in famine years, the $\delta^{18}O$ values were lower than the averages for most segments (Fig. S3), implying that the occurrence of famines in this period is related to wet, rather than dry, atmospheric conditions in the growing season. Indeed, four of the six famine years recorded in $\geq 3$ titles of Town/City history during the 1701–1900 period occurred in the same or following year of long rain with three or more descriptions, while only one occurred in the same, or following, year of drought (Table S2). Similar relationships were found for poor harvest and drought/long-rain years (four of seven poor-harvest years coincided with long-rain years, while one poor-harvest year coincided with a drought year). This suggests that many famines in historical times were attributed to crop failure due to long rain and related climates, such as poor sunshine or low temperature, in growth seasons. Some disaster descriptions supporting this speculation can be found in the Town/City histories.

Insect damage also possibly affect intra-ring $\delta^{18}O$ variation. However, there are only two years (1825 and 1836) when the insect damage was described in $\geq 3$ titles of the Town/City histories in the 1701–1900 period (Table S2). It is an insufficient number of sample years to examine the relationship with tree-ring $\delta^{18}O$. Also, we could not find common anomalous features in intra-ring $\delta^{18}O$ variations in these two years by visual comparison of the graphs. Although it is known that large outbreaks of caterpillar or sawfly affect the tree-ring width and $\delta^{18}O$ (e.g., Huang et al., 2008; Yuri et al., 2014), most of all insect damage recorded in historical documents in Japan occurred by plant hoppers or grasshoppers. Although these insects damage seriously on agricultural crops such as rice plants, they do not eat cedar leaves. Therefore, it is unlikely they affected the growth of the sample tree.

Other possible climatic factors affecting tree-ring $\delta^{18}O$ include spells of abnormal high/low temperature in the growing season. In the Town/City history documents used in this study, significant cold summer years recorded in $\geq 3$ titles in the 1701–1900 period are only two (1836 and 1854), and common anomalous features in intra-ring $\delta^{18}O$ variations in these two years were not found. In Central Japan, low temperature in summer season is usually accompanied by large precipitation (in this case, cold summers in 1836 and 1854 coincided with long rain and flood, respectively, according to the Town/City

histories). Therefore, we decided that it was more reasonable to reflect cold summers in our research using long-rain and flood records that are more common than cold-summer records in the Town/City history documents. Records on abnormal high temperature in spring/summer/autumn are even fewer than cold summer in documentary records used in this study. Also, when we conducted correlation analysis between tree-ring $\delta^{18}O$ of our sample tree and 10-day average temperature at Gifu station, weaker correlation was found than with relative humidity or precipitation in the growing season of the sample tree. In the central part of Japan, $\delta^{18}O$ in precipitation is strongly influenced by altitude (altitude effect) and rainfall intensity (rainfall effect), and the temperature effect is marginal (e.g., Yabusaki & Tase, 2005). Therefore, it is generally difficult to read the signal of temperature variation in tree-ring $\delta^{18}O$ in Central Japan.

### 3.4 Extraction of disaster years using the intra-ring $\delta^{18}O$

Based on the comparison of intra-ring $\delta^{18}O$ variation by disaster types, we attempted to set the threshold of $\delta^{18}O$ value for each segment to extract disaster year to verify and complement the chronology of disaster year reconstructed from documentary disaster records. Hereafter, we focused on drought and long rain from among the eight types of disaster classified in Table 1 because these disaster types are directly related to anomalous hydroclimatic conditions lasting weeks or months that are likely reflected in tree-ring $\delta^{18}O$. Fig. 6 shows the intra-ring $\delta^{18}O$ variations for documentary-based drought and long-rain years in 1701–1900, together with $\delta^{18}O$ deviation of the average for drought/long-rain years from the average for all years in this period.

#### 3.4.1 Drought years

The mean isotopic ratios for the drought years were higher than the all-year average values in later segments. The deviations were +0.71‰, +1.13‰, and +1.36‰ in the 4th, 5th, and 6th segments for the drought years, respectively (Fig. 6). Deviations in earlier segments were small. This means that drought records in historical documents tended to capture dry conditions in the later part of the growing season of the sample tree (roughly June to July), which corresponds to the Baiu rainy season in Central Japan.

To identify drought years corresponding to those extracted from documentary records, we defined a criterion of intra-ring $\delta^{18}O$ values based on the above results. As a result of trial-and-error searches so as to maximize the concordance with documentary-based dought years, the criterion was set as follows: The deviation from the all-year average is larger than +0.60‰ continuously in the 4th and 5th segments or in the 5th and 6th segments. Based on this criterion, 38 drought years were extracted in the 1701–1900 period (Fig. 7a). These extracted drought years included 78% (7 of 9 years) of documentary-based drought years recorded in $\geq$ 3 titles of Town/City history and 86% (6 of 7 years) of those recorded in $\geq$ 4 titles.

#### 3.4.2 Long-rain years

In all segments, the mean $\delta^{18}O$ values for documentary-based long-rain years were lower than the mean values for all years 1701–1900 (Fig. 6). Those deviations were -0.81‰, -0.62‰, -0.53‰, -0.88‰, -0.77‰, and -0.41‰ in the 1st to 6th segments, respectively. These results imply that long-rain records in historical documents capture wet conditions during the whole period of the growing season of the sample tree (roughly April to July).

To identify long-rain years corresponding to those extracted from documentary records, we defined a criterion of intra-ring $\delta^{18}O$ values based on the above results. As a result of trial-and-error searches, the criterion was set as follows: The deviation from the all-year average is smaller than -0.60‰ continuously in any four consecutive segments (that is, 1st to 4th, 2nd to 5th, or 3rd to 6th segments). Based on this criterion, 33 long-rain years were extracted, including 67% (6 of 9 years) and all (3 of 3 years) documentary-based long-rain years recorded in $\geq$ 3 and $\geq$ 4 titles of Town/City history, respectively.

The extraction criteria described above successfully captured most of the major documentary-based drought/long-rain years, but several years could not be captured. It is probably because the season of occurrence of drought/long-rain events recorded in historical documents spans longer period than that of tree-ring $\delta^{18}O$ data. For example, the long rain in 1866 was not extracted by the intra-ring $\delta^{18}O$ values and is recorded in three titles of Town/City history, one of which contains a mention of a long-rain occurrence in October of the solar calendar. It is unlikely that wet conditions after the growing season of the sample tree are recorded in the intra-ring $\delta^{18}O$ data. On the other hand, wet conditions earlier in the season, such as anomalous early onset of the Baiu season, can be captured in the intra-ring $\delta^{18}O$ data.

### 3.4.3 Verification of extraction results

As shown in the analysis in sections 3.4.1 and 3.4.2, the intra-ring $\delta^{18}O$ data and documentary drought/long-rain records used in this study were expected to be closely related to dry/wet conditions in the Baiu season in Central Japan. Herein, we validate the result of drought/long-rain years in the previous subsections by comparing it to the reconstructed annual variation of the length and magnitude of the Baiu season in Mizukoshi (1993). Mizukoshi (1993) estimated the date of the onset and end of the Baiu season for every year since 1751 and the precipitation for the duration from the onset to the end of the Baiu season (here we refer to it as "Baiu duration") for every year since 1692 based on daily weather distribution in Central Japan.

We calculated the mean values of the number of days and the precipitation of Baiu duration for drought and long-rain years recorded in $\geq 1$ to $\geq 4$ titles of Town/City history together with the significance of their difference from the mean values for all years in the analyzed period (Table S3). The mean number of days for drought (long-rain) years was significantly less (more) than the mean for all years 1751–1900 of 37 days in the case of $\geq 3$ and $\geq 4$ ($\geq 2$ to $\geq 4$) titles (at $p < 0.05$). The mean precipitation for drought (long-rain) years was significantly less (more) than the mean for all years 1701–1900 of 351 mm in the case of $\geq 2$ to $\geq 4$ ($\geq 1$ to $\geq 4$) titles (at $p < 0.05$). The highest significance was found in the case of $\geq 3$ titles at $p < 0.01$. Based on this result, we considered the $\geq 3$ titles as the threshold of extracting drought/long-rain years based on disaster descriptions in the Town/City histories.

Similarly, we calculated the mean values of the number of days and the precipitation of Baiu duration for drought and long-rain years extracted by the intra-ring $\delta^{18}O$ data in sections 3.4.1 and 3.4.2, together with the significance of their differences from the mean values for all years in the analyzed period. The mean number of days for drought (long-rain) years was 27 (43) days ($n = 24$ ($n = 28$)), significantly less (more) than the mean for all years 1751–1900 at $p < 0.001$ ($p < 0.05$). The mean precipitation for drought (long-rain) years was 261 (448) mm ($n = 38$ ($n = 33$)), significantly less (more) than the mean for all years 1701–1900 at $p < 0.001$ ($p < 0.01$).

These results suggest that drought (long-rain) years are related to shorter (longer) Baiu duration and less (more) precipitation in the Baiu season, based on both of historical documents and intra-ring $\delta^{18}O$ data.

### 3.4.4 Application to the 17th and 20th centuries

Based on the results in sections 3.4.1 and 3.4.2, we attempted to extract drought and long-rain years using intra-ring $\delta^{18}O$ data for the 1609–1700 and 1901–1949 periods, when there were few disaster records in the Town/City history documents.

For the period 1609–1700, 12-divided intra-ring $\delta^{18}O$ data were converted to 6-divided data by averaging two consecutive segments. 38 drought years and 1 long-rain year were extracted in 1609–1700 period when the values of thresholds described in sections 3.4.1 and 3.4.2 were applied. This imbalance in the number of droughts and long rains is explained by the long-term decreasing trend in tree-ring $\delta^{18}O$, i.e., the mean $\delta^{18}O$ value for 1609–1700 was higher than that for 1701–1900 by approximately 0.8‰ (Fig. 1b). It is unlikely that this decreasing trend is attributed to an "age effect" (or "juvenile effect") in tree-ring $\delta^{18}O$ since the age of the sample tree of this study was older than 250 years in 1609. Although this decreasing trend

in $\delta^{18}$O seems to reflect actual long-term variation in the hydroclimatic conditions, no excessive imbalance in the number of droughts and long rains was found in the documentary records (Table 1). This suggests that the criteria for recording drought/long rain in historical documents were based on the relative deviation from the normal state of the hydroclimatic conditions at that time. Therefore, we set the threshold values for extracting drought and long-rain years in 1609–1700 based on the $\delta^{18}$O deviation from the mean for 1609–1700 for each segment. 19 drought years and 9 long-rain years were extracted using these threshold values.

Similarly, we set the values of thresholds for extracting drought and long-rain years in 1901–1949 based on the $\delta^{18}$O deviation from the mean for 1901–1949 for each segment. The period 1950–2020 was omitted since 6-divided intra-ring $\delta^{18}$O data were available for only 15 years (Fig. 1a). Consequently, 8 drought and 6 long-rain years were extracted using these threshold values.

**3.5 Drought and Long-rain chronologies**

We applied the criteria set as described in the Section 3.4 to all the years for which 6-divided intra-ring $\delta^{18}$O data are available in 1609–1949 and obtained the drought and long-rain chronologies. In Fig. 7, drought and long-rain years extracted from tree-ring $\delta^{18}$O are indicated (short bars) together with those recorded in ≥4 titles of Town/City history (long bars). Documentary-based drought and long-rain years clustered in the period from the late 18th to the mid-19th centuries and were very sparse before the mid-18th century. This observation reflects the lower availability of documentary sources for the 17th century. Documentary sources are also sparse after the late 19th century as the Edo era ended, and the Japanese political system shifted to modernity in 1868.

Focusing on the period 1770–1850, in which major famines Tenmei (1782–1787) and Tempo (1833–1839) had occurred and the documentary sources are relatively rich, agreement of extracted years based on documentary and tree-ring sources was excellent for both long rain and drought years (Fig. 7b). This is probably because most droughts recorded in documentary sources correspond to lack of precipitation in the Baiu rainy season (typically June to mid-July) that is covered by the growing season of the sample tree. Meanwhile, long rains can correspond to both of the following situations: Wet pre-Baiu (typically May) and post-Baiu (late July to August) seasons, of which the latter was not covered by the growing season of the sample tree. The agreement of extracted years from documentary and tree-ring sources may be a better match if a sample tree having a longer growing period, covering the post-Baiu season, is obtained.

Mizukoshi (1993) showed increased precipitation during the Baiu season around 1750, 1780, and 1830–1840. The 1780s and 1830s are known as the periods of major famines in the Edo era; the Tenmei (1782–1787) and Tempo (1833–1839) famines (Mizukoshi,1993; Yamakawa,1993). These famines were caused by crop failure due to prolonged cold and wet summers. Cold summers also occurred during the 1920–1930s, causing a famine in Northeast Japan (Yonenobu and Eckstein, 2006; Kondo, 1988; Hirano et al., 2013). The cold summers of 1931 and 1934 caused severe crop failures that adversely impacted the Japanese economy. Fig. 7a shows several periods when long-rain was concentrated, such as the 1750s, 1780s, and 1920–30s, which suggests that during this period, central Japan experienced cold and wet summers that rivaled the conditions that led to the major famines of the Edo era.

Fig. 7a also shows periods of concentrated drought years, such as the 1700–10s, around 1800, and around 1820. However, no records or data were found to indicate a significant occurrence of famine or disaster in these periods (Table S2) since drought damages could be mitigated by irrigation systems in the early modern age in Japan (Yamakawa, 1993; Takeuchi et al., 1999), in contrast to damages by long rain (low temperature and lack of sunlight).

## 4 Conclusions

In this study, we attempted to construct long-term, temporally consistent chronologies of climate disasters using tree-ring $\delta^{18}O$ and documentary data. Historical documents were useful as sources of information on the occurrence of disasters in historical times in Japan. However, those records are subjective and discrete in time, hindering objective analysis of long-term variation in frequency or magnitude of disasters.

To overcome this problem, we employed the following approaches: First, to collect as many disaster records as possible efficiently instead of using original documents, we referred to published "Town/City histories" comprehensively collected and compiled from historical documents in each municipality. Listing the disaster records in 20 titles of Town/City history in chronological order allowed us to evaluate, to some extent, the magnitude of each disaster event by counting the number of sources (titles) referring to it. Second, we analyzed intra-tree-ring $\delta^{18}O$ to calibrate and complement the documentary-based disaster records. We analyzed intra-ring $\delta^{18}O$ patterns for years of each type of disaster recorded in documentary sources and found that most major documentary-based drought and long-rain years could be captured by setting suitable criteria for intra-ring $\delta^{18}O$ values. Drought and long-rain chronologies spanning four centuries in the Tono area in Central Japan were then obtained by applying these criteria.

The novelty of this study lies in the method used to obtain temporally consistent long-term hydroclimatic data of equivalent quality to historical documents. Historical documentary records offer more detailed information on each disaster event than other proxy data; however, securing long-term continuity remains challenging. Furthermore, the quantity of those records generally decreases for older times. The method used in this study can be applied further back in time with consistent quality as long as adequate wood samples are available and provide useful information to analyze long-term trends, periodicity, amplitude, or the possibility of abrupt change in the magnitude and frequency of hydroclimatic extremes. Such analysis is essential to designing protection from climate disasters unexperienced in recent centuries or adaptation to possible future climate change in long-term perspectives.

The range of $\delta^{18}O$ variation within an annual ring was generally larger than that of inter-annual variation (Fig. 4b), which indicates that substantial information originally contained in tree-ring $\delta^{18}O$ variation is obscured in annually measured tree-ring $\delta^{18}O$ data. Many drought and long-rain years in our resultant chronologies could not be found in annually measured $\delta^{18}O$ data since those are hydroclimatic anomalies in a time scale smaller than the length of the growing season of the sample tree. Moreover, it may be possible to obtain disaster chronologies by season of occurrence, i.e., earlier and later in the growing season separately, though further accumulation of measured data and improvement in analyzing techniques remains warranted. Meanwhile, another difficulty arises in intra-annual tree-ring analyses due to the lack of a distinct time marker like a ring boundary within an annual ring. In this study, we estimated the season corresponding to each segment within an annual ring by correlating the $\delta^{18}O$ values with 10-day relative humidity and precipitation data for the period of instrumental observation. However, substantial ambiguity remains challenging. Developing a method of more precise dating within an annual ring remains warranted.

**Data availability**

**Supplement link**: the link to the supplement will be included by Copernicus, if applicable.

**Author contributions:** HI, KS designed the research and wrote the paper; HI, KS, YK, TN collected tree-ring samples; HI, KS extracted cellulose from tree-rings; ZL, MS measured isotope ratios of tree-ring cellulose; HI, KS, YK analyzed isotopic data mathematically. All authors discussed the results and contributed to the manuscript.

**Competing interests**: The authors declare that they have no conflict of interest.

**Acknowledgments**: We would like to thank the Ookute Community Promotion Council for providing the tree-ring sample. And we would like to thank for the time and efforts three anonymous referees in editing. This work was supported by JSPS KAKENHI Grant Numbers JP21H04980, JP22H04938, and JP23K00974.

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

**Figures**

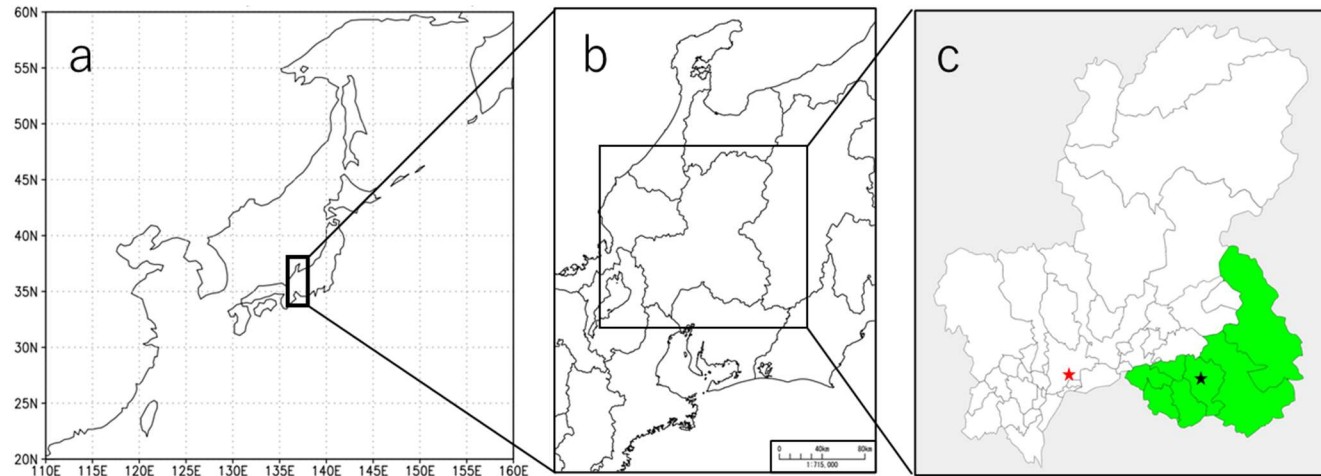

Fig. 1 Maps showing the location of the study site: (a) Central Japan. (b) Gifu Prefecture. (c) Tree-ring sampling site (black star), meteorological station (red star), and Tono area where Town/City histories were researched (green area).

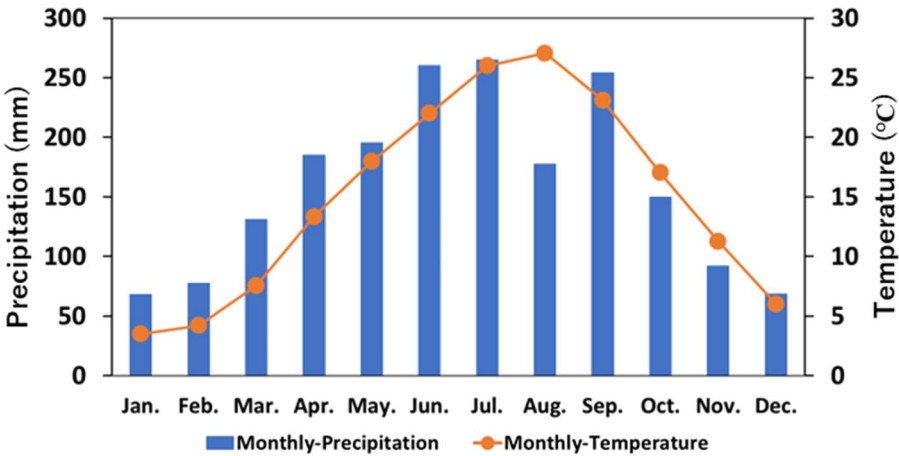

Fig. 2 Climatological data: Average monthly temperature and precipitation at Gifu Local Meteorological Office (1883–2020).

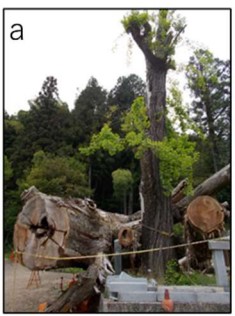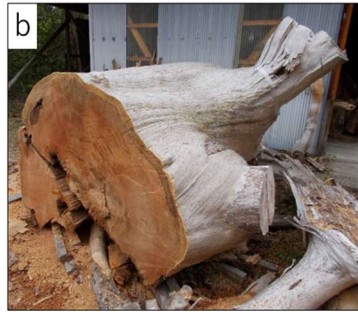

Fig. 3 Sample tree: Japanese cedar (*Cryptomeria japonica*). (a) The fallen trunk at Okute-Shinmei Shrine. (b) The cross section

where the disk samples were taken.

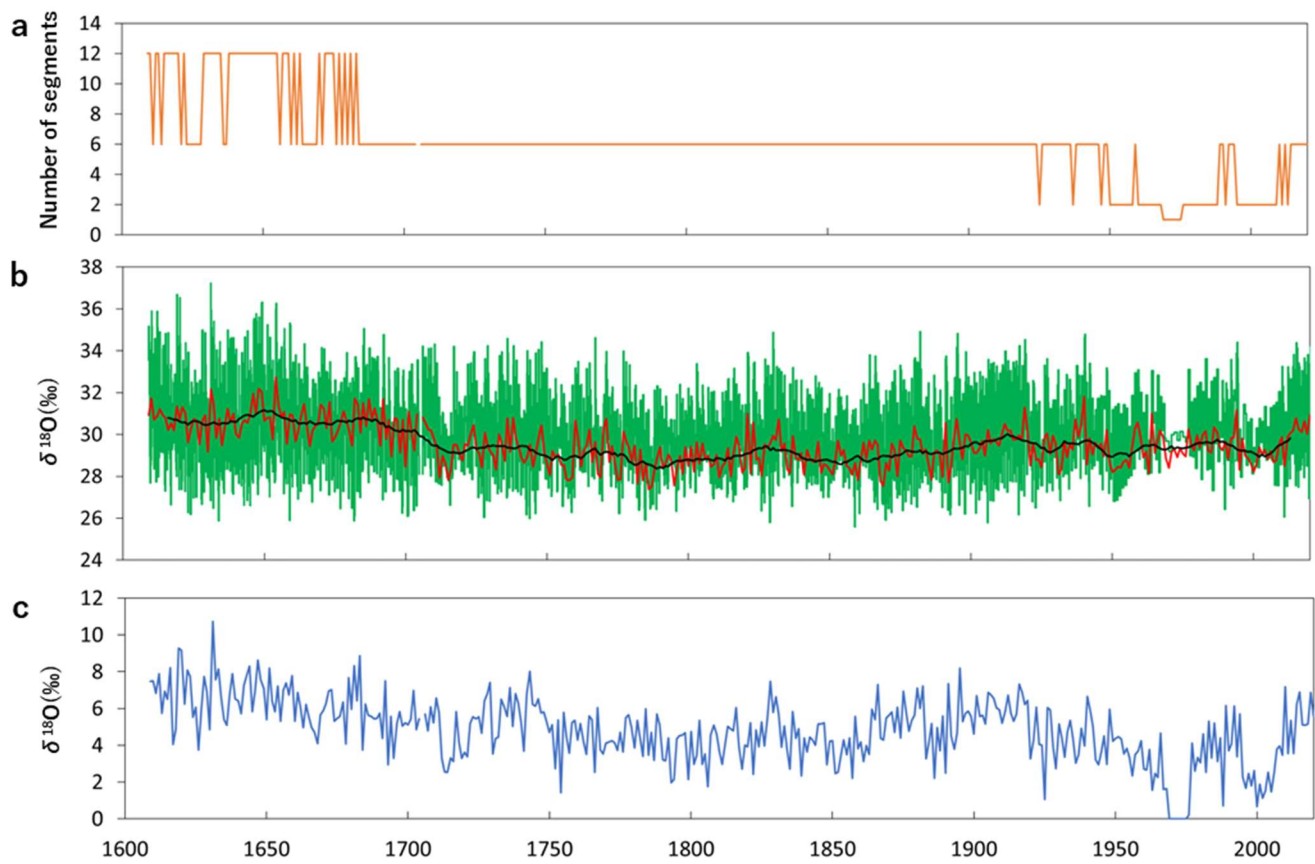

Fig. 4 Time-series graphs showing the results of δ18O measurement (1609–2020). (a) Number of segments in each annual ring. (b) Originally measured intra-ring δ18O values (green line), annual average (red), and 15-year moving average (black). (c) Range of variation (maximum minus minimum) of δ18O within each annual ring.

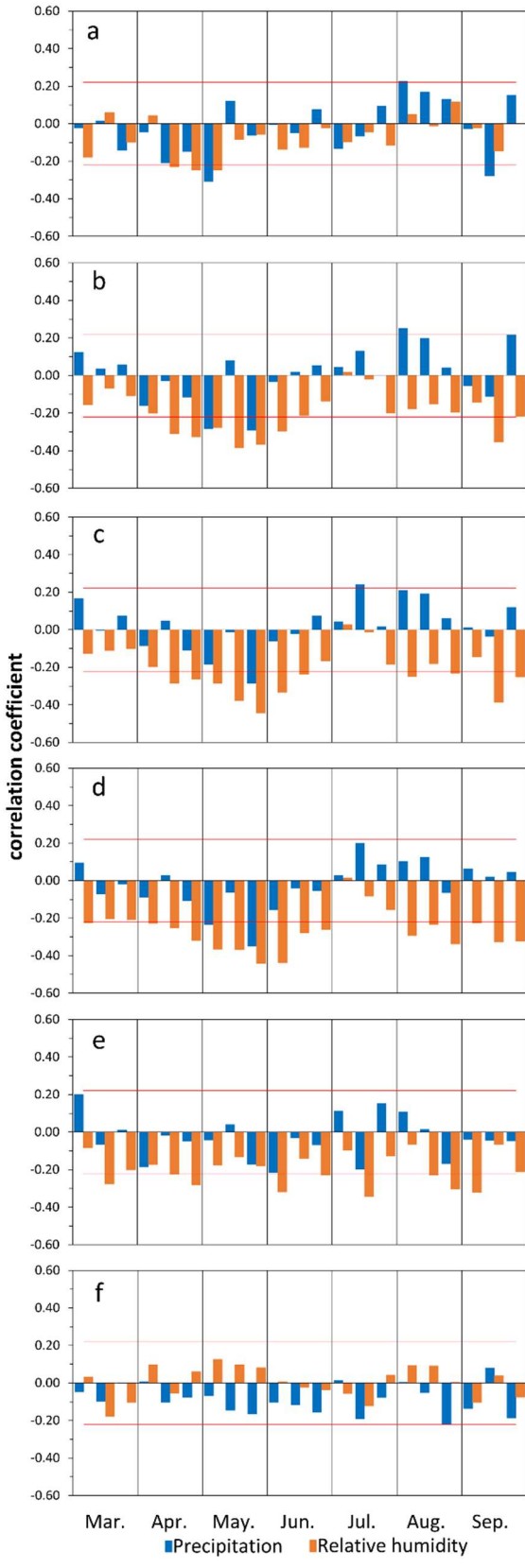

Fig. 5 Correlation between δ¹⁸O values and observed 10-day precipitation (blue bars) and mean relative humidity (orange) for each of (a) segment 1 to (f) segment 6. Horizontal red lines in each graph indicate a significance level of $p < 0.05$. The sample size (number of rings for which the number of segments is six in the observation period 1883–2020) is 74. Meteorological data were observed at Gifu Local Meteorological Office (136°45'45" E, 35°25'58" N, 12.7 m a.s.l.).

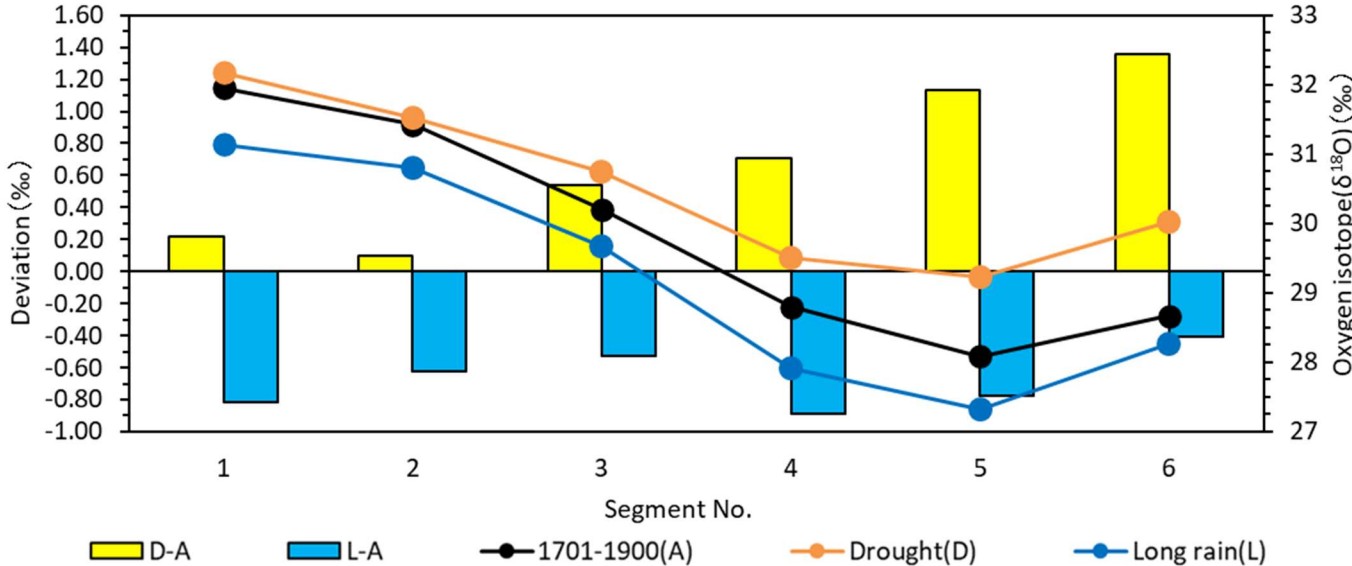

Fig. 6 Mean value of $\delta^{18}O$ in each segment for the documentary-based drought years (orange dots, 9 years in total), long-rain years (blue dots, 9 years), and 1701–1900 (black dots, 200 years) together with their deviation from the 1701–1900 average (yellow and blue bars, drought and long-rain years, respectively). Yellow bars (blue bars) indicate the deviation of the average $\delta^{18}O$ for drought (long-rain) years from the average $\delta^{18}O$ for all years (1701–1900).

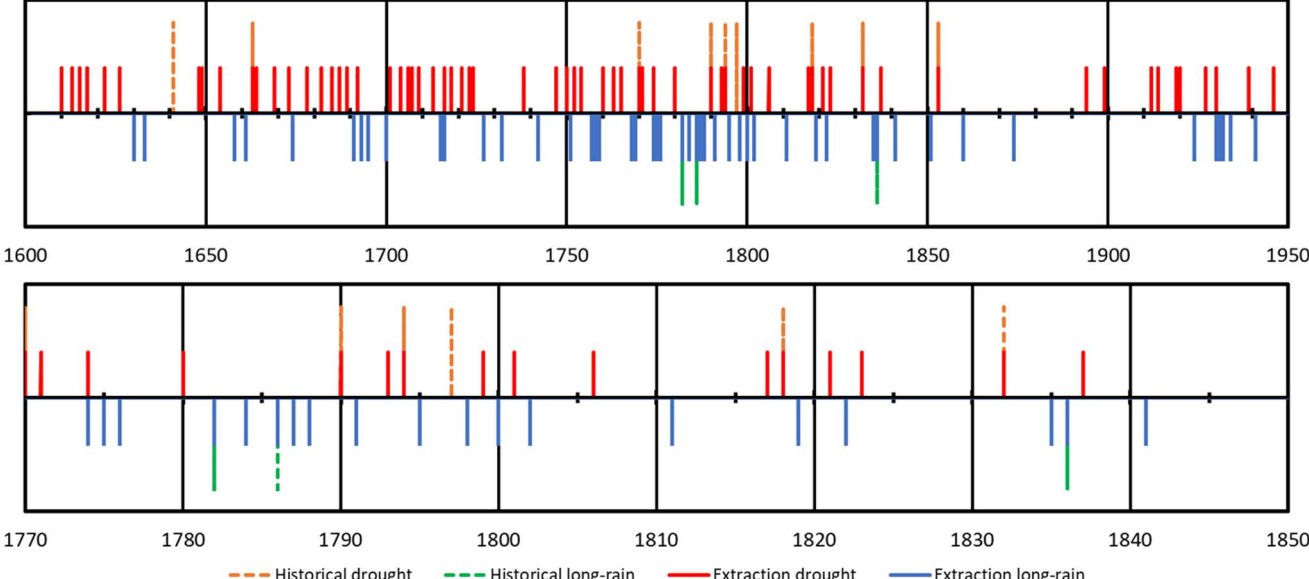

Fig. 7 Time-series graphs showing the years of drought and long-rain extracted from tree-ring $\delta^{18}$O and documentary data. Historical drought (orange bars): Documentary-based drought years; Historical long-rain (green): Documentary-based long-rain years; Extraction drought (red bars): $\delta^{18}$O-based drought years; Extraction long-rain (blue bars): $\delta^{18}$O-based long-rain years. Historical drought and Historical long-rain represent drought and long-rain years recorded in ≥ 4 titles of Town/City history. The lower graph is enlargement for the 1770–1850 period of the upper graph.

**Tables**

Table 1 Total number of disaster descriptions in a total of 20 titles of Town/City history in the Tono area for each of eight disaster types for each 50-year interval in 1601–1900.

| | Long rain | Drought | Flood | Heavy rain | Poor harvest | Famine | Heavy wind | Insect damage |
|---|---|---|---|---|---|---|---|---|
| 1601–1650 | 2 | 6 | 14 | 1 | 6 | 12 | 0 | 0 |
| 1651–1700 | 4 | 8 | 22 | 2 | 4 | 12 | 0 | 3 |
| 1701–1750 | 3 | 9 | 17 | 2 | 9 | 5 | 1 | 2 |
| 1751–1800 | 28 | 44 | 27 | 12 | 26 | 18 | 3 | 0 |
| 1801–1850 | 12 | 29 | 16 | 9 | 29 | 24 | 15 | 11 |
| 1851–1900 | 10 | 17 | 32 | 10 | 11 | 5 | 6 | 1 |
| 1701–1900 | 90% | 88% | 72% | 92% | 88% | 68% | 100% | 82% |

The bottom row shows the percentage of the number of disaster descriptions in 1701–1900 relative to the total period 1601–1900.