# Peer review of "Reconstruction of drought and long-rain chronologies since the 17th century in Central Japan using intra-annual tree-ring oxygen isotope ratios and documentary records"

_EGUsphere, 2024_

## Author Response (AR2)

23 September 2024

Dear Editor:

I wish to submit the re-revised manuscript titled "**Reconstruction of drought and long-rain chronologies since the 17th century in Central Japan using intra-annual tree-ring oxygen isotope ratios and documentary records**." The manuscript ID is EGUSPHERE-2024-627.

We thank you and the reviewers for your thoughtful suggestions and insights. The manuscript has benefited from these insightful suggestions. We look forward to working with you and the reviewers to move this manuscript closer to publication in the *Climate of the Past.*

The manuscript has been rechecked and the necessary changes have been made in accordance with the reviewers' and the editor's suggestions. The responses to all comments have been prepared and attached herewith. We inserted mentions of relationships between intra-ring $\delta^{18}O$ variation in our sample tree and anomalous temperature and insect damage to the end of the section 3.3 (lines 178-198 in the re-revised manuscript) in accordance with the editor's suggestions.

Notes: Line numbers in the replies to the reviewers and the editor correspond to those indicated in the preprint. Revised texts shown in the replies according to the reviewers' and the editor's comments were marginally changed after submission to the interactive discussion according to proofreading.

Thank you for your consideration. We look forward to hearing from you.

Sincerely,

庄建治朗

Kenjiro Sho

Department of Architecture,
Civil Engineering, and Industrial Management Engineering,
Nagoya Institute of Technology,
Nagoya 466-8555, Japan
Phone number; +81-52-735-5494
Email address; show@nitech.ac.jp

Thank you very much for providing valuable important comments. We are thankful for the time and efforts you expended. Our responses to the RC1 comments are as follows:

1. *It would be much better that the authors add a figure showing the comparison between the two chronologies.*

Reply: We add the graph as Fig. S1 below according to the suggestion. The current Fig. S1 will be re-titled as Fig. S2. In light of these changes, Lines 118-122 are reviesed as follows.

Lines 118-122:

Nakatsuka et al. (2020) conducted a study on the long-term variation of the climatological component of tree-ring $\delta^{18}O$ in Central Japan. Their resultant chronology was compared with the climatological component of annual-averaged tree-ring $\delta^{18}O$ of the sample tree of this study calculated by folloing the method of Nakatsuka et al. (2020) (Fig. S1). These two chronologies showed similar long-term variations for almost the entire measurement period of this study. A significant positive correlation was also observed, r = 0.52 (n = 395, p < 0.01).

[Figure]

Fig. S1 Time-series graph of the climatological component of annual-averaged tree-ring $\delta^{18}O$ of the sample tree of this study (blue line) and the central Japan master chronology (Nakatsuka et al., 2020, orange line)

2. _Lines 149–150: the authors wrote: the 4th to 6th segments in drought and 1st and 5th segments in long rain were significant (p<0.05), and so was the 4th segment in long rain (p<0.10). It is hard to understand._

Reply: We tested the significance of the difference between $\delta^{18}O$ average for drought/long-rain years and for the all years 1701–1900 for each segment. The results showed that significant differences were expressed in the 4th, 5th, and 6th segments for drought years. Significant differences were expressed in segments 1st 4th, and 5th for long-rain years.

    Following the comment, Lines 148-150 are revised as follows.

Lines 148-150:

The $\delta^{18}O$ values were generally higher in drought years and lower in long-rain and flood years than the average of all years for 1701–1900 (Fig. S2), especially, the $\delta^{18}O$ average in drought years were significantly different from $\delta^{18}O$ average during 1701–1900 for 4th to 6th segments (p < 0.05). $\delta^{18}O$ average in long-rain years were significantly different from $\delta^{18}O$ average during 1701–1900 for 1st and 5th segments (p < 0.05), and the 4th segment (p < 0.10).

3. _Lines 204-205: It is not very clear how the mean precipitation for drought and long-rain years extracted by the intra-ring $\delta^{18}O$ data in 1701–1900 are reconstructed._

Reply: We used the reconstructed precipitation during the Baiu duration by Mizukoshi (1993). According to this reconstructed precipitation, we calculated the mean precipitation for the drought and long-rain years extracted by intra-ring $\delta^{18}O$ in this study for 1701–1900.

    Based on the above, Lines 204-205 are revised as follows.

Lines 204-205:

Similarly, the mean of precipitation reconstructed by Mizukoshi(1993) for drought and long-rain years extracted by the intra-ring $\delta^{18}O$ data (drought years extracted in 3.4.1 (38 years), long-rain years extracted in 3.4.2 (33 years)) in 1701–1900.

4. _From Figure 4a, it is clear that during 1700-1900 AD the annual ring is divided into 6 segments for each year. So, I would suggest the authors clarify the variation of number of segments clearer in the text and figure captions._

Reply: According to the suggestion, we add the explanation of the variation of number of segments to lines 88-90.

Lines 88-90:

We divided…several years. Each ring was devided into 12 segments for 50 years of 1609–1683 period, and into 2 segments for 52 years of 1925–2013 period. We measured annually for 7 years 1969–1975 because the ring widths were too narrow (Fig. 4a).…

*5. Lines 220-224: it should be moved to Section 3.4?*

Reply: Following the comments, we move lines 210-223 to the end of Section 3.4 as '3.4.4 Application to the 17th and 20th centuries'. The text is also revised as follows.

**3.4.4 Application to the 17th and 20th centuries**

Based on the results in 3.4.1 and 3.4.2, we attempt to extract drought and long-rain years using intra-ring $\delta^{18}O$ data for 1609–1700 and 1901–1949 periods, when there are few disaster records in the Town/City history documents.

For the period 1609–1700, 12-divided intra-ring $\delta^{18}O$ data were converted to 6-divided data by averaging two consecutive segments. 38 drought years and 1 long-rain year were extracted during 1609–1700 when the values of thresholds described in Sections 3.4.1 and 3.4.2 were applied. This imbalance in the number of droughts and long rains is explained by the long-term decreasing trend in tree-ring $\delta^{18}O$. That is, the mean $\delta^{18}O$ value for 1609–1700 is higher than the mean for 1701–1900 by approximately 0.8‰ (Fig. 1b). It is unlikely that this decreasing trend is an "age effect" (or "juvenile effect") in tree-ring $\delta^{18}O$ since the age of the sample tree of this study was already older than 250 years in 1609. Although this decreasing trend in $\delta^{18}O$ seems to reflect actual long-term variation in the hydroclimatic conditions, excessive imbalance in the number of droughts and long rains was not seen in the documentary records (Table 1). This suggests that the criteria for recording drought/long rain in historical documents were based on relative deviation from the normal state of the hydroclimatic conditions at that time. Therefore, we set the values of thresholds for extracting drought and long-rain years in 1609–1700 based on the $\delta^{18}O$ deviation from the mean for 1609–1700 for each segment. 19 drought years and 9 long-rain years were extracted by using these threshold values.

Similarly, we set the values of thresholds for extracting drought and long-rain years in 1901–1949 based on the $\delta^{18}O$ deviation from the mean for 1901–1949 for each segment. 1950–2020 was omitted since 6-divided intra-ring $\delta^{18}O$ data were available for only 15 years in this period (Fig. 1a). Consequently, 8 drought years and 6 long-rain years were extracted using these threshold values.

**3.5 Drought and Long-rain chronologies**

We applied the criteria set as described in the Section 3.4 to all the years for which 6-divided intra-ring $\delta^{18}O$ data are available in 1609–1949 and obtained the drought and long-rain chronologies. In Fig. 7, drought and long-rain years extracted from tree-ring $\delta^{18}O$ are indicated (short bars) together with those recorded in $\geq$ 4 titles of Town/City history (long bars).

6.  *Figure 6: How is the deviation calculated?*

Reply: Deviation is the difference between the average $\delta^{18}O$ for drought/long-rain years and the average $\delta^{18}O$ for all years 1701-1900. We revise lines 159-160 and the caption of Fig. 6 as follows.

Lines 159-160:

Fig. 6 shows the intra-ring $\delta^{18}O$ variations for documentary-based drought and long-rain years in 1701–1900 together with $\delta^{18}O$ deviation of the average for drought/long-rain years from the average for all years 1701–1900.

Fig. 6 caption:

Fig. 6 Mean value of $\delta^{18}O$ in each segment for the documentary-based drought years (orange dots, 9 years in total), long-rain years (blue dots, 9 years), and the all years 1701–1900 (black dots, 200 years) together with their deviation from the 1701–1900 average (yellow bars and blue bars for drought and long-rain years, respectively). Yellow bars (blue bars) are deviation of the average $\delta^{18}O$ for drought (long-rain) years from the average $\delta^{18}O$ for all years 1701–1900.

7.  *The authors wrote: The mean value and standard deviation of $\delta^{18}O$ for each segment for each of the years of long rain, drought, flood, and famine are shown in Fig. S1. However, I did not find any content with regard to standard deviation.*

Reply: It is our editing mistake. We delete "and standard deviation". Also, we renumber Fig. S1 to Fig. S2 (according to addition of Fig. S1 in comment #1).

Fig. S2 caption:

Fig. S2 Intra-ring $\delta^{18}O$ variation by disaster types. The mean of $\delta^{18}O$ …

Thank you very much for providing valuable important comments. We are thankful for the time and efforts you expended. Our responses to the RC2 comments are as follows:

*1. Apart from the records about precipitation, the documentary records about temperature could also have affected $\delta^{18}O$ of tree ring, e.g., abnormally hot or cool growing seasons (including summer, spring and autumn), cold waves in growing seasons (unseasonal frosts and snow).*

Reply: In the Town/City history documents used in this study, the number of records on cold summer is limited, 5 years (1703, 1736, 1787, 1836, and 1854) in 1701–1900 period. Significant cold summer years recorded in more than three titles of the Town/City histories are only two (1836 and 1854). Also, most of these cold summers coincided with long rain or flood. There are more than three titles of the Town/City histories recording long rain in 1836 and flood in 1854. Therefore, we decided that it was more reasonable to reflect cold summers in our research using long-rain and flood records that are more common than cold-summer records in the Town/City history documents. Records on abnormally hot spring/summer/autumn or cold waves are even fewer in documentary records used in this study.

   Also, when the correlation between tree-ring $\delta^{18}O$ of our sample tree and 10-day average temperature was calculated, we found lower correlation than relative humidity or precipitation in the growing season of the sample tree. In the central part of Japan, $\delta^{18}O$ of precipitation is strongly influenced by altitude (altitude effect) and rainfall intensity (rainfall effect), and the temperature effect is marginal (e.g., Yabusaki & Tase, 2005). Therefore, it is generally difficult to read the signal of temperature variation in tree-ring $\delta^{18}O$ in Central Japan.

References:

Yabusaki, S., and Tase, N.: Characteristics of the $\delta^{18}O$ and $\delta D$ of Monthly and Event Precipitation in Tsukuba from 2000 to 2002, J. Jpn Soc. Hydrol. Water Resour., https://doi.org/10.3178/jjshwr.18.592, 2005.

*2. Insect damage is another type of factor; it would be better to analyze it separately.*

Reply: There are only two years (1825 and 1836) when the insect damage was described in more than three titles of the Town/City histories in the 1701–1900 period (Table S2). It is an insufficient number of sample years to examine the relationship with tree-ring $\delta^{18}O$. We conducted the analysis described in 3.3 excluding these two years, but the result was nearly the same as the original result. Although it is known that large outbreaks of caterpillar or sawfly affect the tree-ring width and $\delta^{18}O$ (e.g., Huang et al., 2008; Yuri et

al., 2014), it has not been reported that these outbreaks occurred in the past in Central Japan. In Japan, most of all insect damage recorded in historical documents occurred by plant hoppers or grasshoppers. These insects damage seriously on agricultural crops such as rice plants, but they do not eat cedar reefs, so they do not affect the growth of the sample tree. Therefore, we could not consider insect damage in this study.

References:

Huang, J.-G., Tardif, J., Denneler, B., Bergeron, Y., Berninger, F.: Tree-ring evidence extends the historic northern range limit of severe defoliation by insects in the aspen stands of western Quebec, Canada, Canadian Journal of Forest Research, 38(9), 2535-2544, https://doi.org/10.1139/X08-080, 2008.

Yuri, G., Federica, C., Nicola La P., Marco, C., Andrea, B.: Tree rings and stable isotopes reveal the tree-history prior to insect defoliation on Norway spruce (*Picea abies* (L.) Karst.), Forest Ecology and Management, 319, 99-106, https://doi.org/10.1016/j.foreco.2014.02.009, 2014.

*3. Sometimes, local environmental change and incidents in the vicinity of the sampling sites could affect the tree rings. It would be better to exclude these effects.*

Reply: We did not find any record (document or other evidence) suggesting significant local environmental change or incident that can affect tree-ring $\delta^{18}O$ of the sample tree within the analyzed period. The sampling site has been managed as a shrine from the beginning of the analyzed period (1600s) to the present. The sample tree always occupied the tree canopy because it was known as a prominent giant tree since the Edo era (17th-19th centuries). Since the sampling site is far from the urban area, it is also free from the effect of localized temperature increases such as the heat-island effect. These facts indicate that the local environment vicinity of the sample tree is relatively constant during the growth period.

**Citation**: https://doi.org/10.5194/egusphere-2024-627-AC2

Thank you very much for providing valuable important comments. We are thankful for the time and efforts you expended. Our responses to the RC3 comments are as follows:

*1. Please provide the necessary details of the historical documents. For example, where the 20 cities/towns are actually located? How is a disaster defined and recorded (by examples)? What is the difference and connection between long rain and flood? Are there descriptions of connection between drought/long rain with famine in the original record?*

Reply:

We add a map showing the location of each municipality where Town/City histories were used in this study as Fig. S1 below according to the suggestion. Fig. S1 in reply to RC1 comment #1 will be re-titled as Fig. S2. Fig. S1 in the original manuscript will be re-titled as Fig. S3. Line 107 is revised as follows.

Line 107:

All 20 titles of Town/City histories in the Tono area were used in this study (Table S1 and Fig. S1).

[Figure]

Fig. S1 Map of Gifu Prefecture showing the location of the municipalities of the Town/City histories used in this study (red solid circles) together with the location of the sample tree (black star) and the meteorological station (red star). The numbers within the red circles link to the numbers in Table S1. We collected 20 Town/City histories, 19 of which were issued by municipalities in the vicinity of the sampling site and one by Gifu Prefecture.

Also, according to the comment, we add more detailed explanations of the historical documentary records to line 106 and lines 151-153.

Line 106:

… documents left in the town/city. Those descriptions of disaster in historical documents are subjective depending on personality and situation of the writer. Most of those descriptions in Town/City histories report just the occurrence of the disaster, while some descriptions contain more detailed information such as the season and duration of the occurrence. Also, continuity of those records is not guaranteed in many cases. Lack of disaster records in a certain year does not necessarily mean non-occurrence of significant disaster in that year. There is another possibility that the disaster record is simply missed for that year. In this study, we collected as many disaster records as possible from Town/City histories in the vicinity area of the sample tree to evaluate the magnitude of disasters quantitatively.

Lines 151-153:

This is consistent with the negative correlation between tree-ring $\delta^{18}O$ and relative humidity (precipitation) in the growing season. Intra-ring $\delta^{18}O$ variation for long-rain and flood years are similar (Fig. S3) because they are both caused by more precipitation than normal. However, they did not necessarily coincide. None of the 9 long-rain years and 6 flood years recorded in $\geq 3$ titles of Town/City history in the 1701–1900 period occurred in the same year (Table S2). This is probably because long rains and floods are associated with rainfall events in different time scales.

$\delta^{18}O$ values in famine years were lower than the averages for most segments (Fig. S3), implying the occurrence of famines in this period can be related to wet atmospheric conditions rather than dry conditions in the growing season. In fact, 4 of 6 famine years recorded in $\geq 3$ titles of Town/City history in the 1701–1900 period occurred in the same or the next year of long rain with $\geq 3$ descriptions, while only 1 famine year occurred in the same or the next year of drought (Table S2). Similar relationships were found for poor-harvest and drought/long-rain years (4 of 7 poor-harvest years coincided with long-rain years, while 1 poor-harvest year coincided with a drought year). This suggests that many famines in historical times were caused by crop failure due to long rain and related climates such as poor sunshine or low temperature in the growth season. Some disaster descriptions supporting this speculation can be found in the Town/City histories.

*2. The quality and reliability of the historical documents needs brief validation, including its comparison to documents from other regions, and application in previous investigation.*

Reply:

We calculated the mean number of days and the mean precipitation of the Baiu season in historical times reconstructed by Mizukoshi (1993) for drought and long-rain years recorded in ≥ 1 to ≥ 4 titles of the Town/City histories. The significance of difference between the mean value of extracted drought/long-rain years and that of the whole period (1751–1900 for the number of days or 1701–1900 for precipitation) is shown in Table S3. According to this result, the highest significance with $p<0.01$ was found for ≥ 3 titles.

Following the comment, we add Table S3 and revise 3.4.3 as follows. Lines 145-146 are revised as follows.

Table S3 The mean numbers of days and the mean precipitation of Baiu duration for drought/long-rain years extracted by disaster records in the Town/City histories together with the number of disaster years extracted (n) and the significance level (p-value) of the difference between the mean values for disaster years and all years in the analyzed period.

| Number of disaster descriptions | The mean number of days (37, 1751-1900) | | | | The mean precipitation (mm) (351, 1701-1900) | | | |
|---|---|---|---|---|---|---|---|---|
| | ≧ 1 | ≧ 2 | ≧ 3 | ≧ 4 | ≧ 1 | ≧ 2 | ≧ 3 | ≧ 4 |
| Drought | 35 (n=49) (p>0.05) | 32 (n=20) (p>0.05) | 27 (n=9) (p<0.01) | 27 (n=7) (p<0.05) | 315 (n=57) (p>0.05) | 265 (n=21) (p<0.05) | 193 (n=9) (p<0.001) | 193 (n=7) (p<0.01) |
| Long-rain | 42 (n=19) (p>0.05) | 45 (n=13) (p<0.05) | 48 (n=9) (p<0.001) | 51 (n=3) (p<0.05) | 475 (n=22) (p<0.01) | 473 (n=13) (p<0.01) | 492 (n=9) (p<0.01) | 645 (n=3) (p<0.05) |

3.4.3:

As shown in the analysis in 3.4.1 and 3.4.2, the intra-ring $\delta^{18}$O data and the documentary drought/long-rain records used in this study are expected to be closely related to dry/wet conditions in the Baiu season in Central Japan. Here we validate the result of drought/long-rain years in the previous subsections by comparing to the reconstructed annual variation of the length and magnitude of the Baiu season in Mizukoshi (1993). Mizukoshi (1993) estimated the date of the onset and end of the Baiu season for every year since 1751 and the precipitation for the duration from the onset to the end of the Baiu season (here we call as "Baiu duration") for every year since 1692 based on daily weather distribution in Central Japan.

We calculated the mean values of the number of days and the precipitation of Baiu duration for drought and long-rain years recorded in ≥ 1 to ≥ 4 titles of Town/City history together with the significance of their difference from the mean values for all years in the analyzed period (Table S3). The mean number of days for drought (long-rain) years was significantly less (more) than the mean for all years 1751–1900 of 37 days in the case of ≥ 3 and ≥ 4 (≥ 2 to ≥ 4) titles at $p<0.05$ or smaller. The mean precipitation for drought (long-rain) years was significantly less (more) than the mean for all years 1701–1900 of 351 mm in the case of ≥ 2 to ≥ 4 (≥ 1 to ≥ 4) titles at $p<0.05$ or smaller. The highest significance was found in the case of ≥ 3 titles at $p<0.01$ or smaller. According to this result, we decided ≥ 3 titles as the threshold of extracting drought/long-rain years based on disaster descriptions in the Town/City histories.

Similarly, we calculated the mean values of the number of days and the precipitation of Baiu duration for drought and long-rain years extracted by the intra-ring $\delta^{18}$O data in 3.4.1 and 3.4.2 together with the significance of their difference from the mean values for all years in the analyzed period. The mean number of days for drought (long-rain) years was 27 (43) days (n=24 (n=28)). This is significantly less (more) than the mean for all years 1751–1900 at $p<0.0001$ ($p<0.05$). The mean precipitation for drought (long-rain) years was 261 (448) mm (n=38 (n=33)). This is significantly less (more) than the mean

for all years 1701–1900 at p<0.001 (p<0.01).

These results suggest that drought (long-rain) years are related to shorter (longer) Baiu duration and less (more) precipitation in the Baiu season in terms of both historical documents and intra-ring $\delta^{18}O$ data.

Lines 145-146:

Herein, years of each type of disaster were extracted as years for which the disaster record is found in ≥3 titles out of 20 Town/City histories. The reason for using ≥ 3 titles as the threshold is described in 3.4.3.

*3. Please provide the scientific rationale of dividing the annual tree-ring sample into six subsamples. The same applies to the association between the $\delta^{18}O$ results in the subsamples and the 10-day relative humidity and precipitation.*

Reply:

According to the suggestion, we add the explanation of the number of segments to lines 88-89 in the original manuscript. Lines 129-131 are revised as follows.

Lines 88-89:

We divided each annual ring into six or more segments of the same width from 1609 to 1949 except for several years. Tree-ring widths of this sample were continuously more than 1 mm in most years until 1949. Six is the practical limit of dividing a 1-mm ring width with our current technique, so we divided each annual ring into six segments as long as possible.

Lines 129-131:

$\delta^{18}O$ measurements were correlated with relative humidity and precipitation data observed at the nearest meteorological station from the sampling site (Gifu Local Meteorological Office, 48 km to the west of the sampling site, Fig. 1c). The growing season of cedar trees in Central Japan is reported as May-August (Hirano et al., 2020). Since monthly meteorological data may be too coarse to analyze correlations with $\delta^{18}O$ data corresponding to six separate periods of the growing season, we used 10-day data that are commonly used as sub-monthly meteorological statistics in Japan.

Fig. 5 shows correlation coefficients between intra-ring $\delta^{18}O$ values and 10-day relative humidity and precipitation for each segment for years when 6-divided $\delta^{18}O$ data are available within the observation period since 1883.

*4. The scientific significance of this study needs to be made clearer. For sample, the authors mention in the conclusion that*
"The range of $\delta^{18}O$ variation within an annual ring was generally larger than that of inter-annual variation (Fig. 4b), which indicates that substantial information originally contained in tree-ring $\delta^{18}O$ variation is obscured in annually-measured tree ring $\delta^{18}O$ data."
*Then what contribution does this study offer if its drought/long rain chronologies also only have annual resolution?*

Reply:

The resultant drought and long-rain chronologies in this study are in annual resolution, but these would not be obtained from tree-ring $\delta^{18}O$ data in annual resolution. Intra-ring $\delta^{18}O$ data allowed us to pick up drought and long -rain years equivalent to those recorded in historical documents. Moreover, as a future possibility, the season of occurrence of disasters may be determined using intra-ring $\delta^{18}O$ data.

To make these points clearer, we insert sentences to line 265 as follows.

Line 265:

… ring $\delta^{18}O$ data. Many of drought and long-rain years in our resultant chronologies could not be found in annually-measured $\delta^{18}O$ data because those are hydroclimatic anomalies in a time scale smaller than the length of the growing season of the sample tree. Moreover, it may be possible to obtain disaster chronologies by season of occurrence, that is, earlier and later in the growing season separately, though further accumulation of measured data and improvement in analyzing technique is needed.

*5. The conclusion would benefit significantly from including a paragraph discussion the implication of its results, for example, the drought/long rain chronologies in the broad field of climate research.*

Reply:

Following the suggestion, we insert a paragraph describing the contributions of the results of this study to relating fields of research between lines 262 and 263.

Between lines 262 and 263:

The significance of this study is that the method to obtain temporally consistent long-term hydroclimatic data of equivalent quality to historical documents was presented. Historical documentary records can offer more detailed information on each disaster event than other proxy data, but it is difficult to secure long-term continuity. Also, the quantity of those records generally decreases rapidly in older times. In contrast, the method of this study can be applied further back in time with consistent quality as long as adequate wood samples are available. It would offer useful information to analyze long-term trends, periodicity, amplitude, or possibility of abrupt change in magnitude and frequency of hydroclimatic extremes. Such analysis is essential to design protection from climate disasters unexperienced in recent centuries or adaptation to possible future climate change in long-term perspectives.

**Reply to the editor**

Thank you very much for providing valuable important comments. We are thankful for the time and efforts you expended. Our responses to the editor's comments are as follows:

*1. R2 asks you to consider temperate documentary records, and I note you have addressed the limited availability of these in your reply to the reviewer, but can I ask you to please summarise your response in the paper as well please.*

*2. Similarly, with regard to your reply on insect outbreaks -have you added this statement to the revised manuscript?*

*3. I would additionally like to suggest that, even with limited data, it is important to note if the years with temperature and insect outbreak anomalies are outliers in your time series. Are they?*

Reply:

According to the suggestions, we add the explanation of intra-riing $\delta^{18}O$ variation for years of insect damage and anomalous high/low temperature in the documentary records to the end of section 3.3 as follows.

End of section 3.3:

Insect damage also possibly affect intra-ring $\delta^{18}O$ variation. However, there are only two years (1825 and 1836) when the insect damage was described in $\geq 3$ titles of the Town/City histories in the 1701–1900 period (Table S2). It is an insufficient number of sample years to examine the relationship with tree-ring $\delta^{18}O$. Also, we could not find common anomalous features in intra-ring $\delta^{18}O$ variations in these two years by visual comparison of the graphs. Although it is known that large outbreaks of caterpillar or sawfly affect the tree-ring width and $\delta^{18}O$ (e.g., Huang et al., 2008; Yuri et al., 2014), most of all insect damage recorded in historical documents in Japan occurred by plant hoppers or grasshoppers. Although these insects damage seriously on agricultural crops such as rice plants, they do not eat cedar leaves. Therefore, it is unlikely they affected the growth of the sample tree.

Other possible climatic factors affecting tree-ring $\delta^{18}O$ include spells of abnormal high/low temperature in the growing season. In the Town/City history documents used in this study, significant cold summer years recorded in $\geq 3$ titles in the 1701–1900 period are only two (1836 and 1854), and common anomalous features in intra-ring $\delta^{18}O$ variations in these two years were not found. In Central Japan, low temperature in summer season is usually accompanied by large precipitation (in this case, cold summers in 1836 and 1854 coincided with long rain and flood, respectively, according to the Town/City histories). Therefore, we decided that it was more reasonable to reflect cold summers in our research using long-rain and flood records that are more common than cold-summer records in the Town/City history documents. Records on abnormal high temperature in spring/summer/autumn are even fewer than cold summer in documentary records used in this study. In the central part of Japan, $\delta^{18}O$ in precipitation is strongly influenced by altitude (altitude effect) and rainfall intensity (rainfall effect), and the temperature effect is marginal (e.g., Yabusaki & Tase, 2005). Therefore, it is generally difficult to read the signal of temperature variation in tree-ring $\delta^{18}O$ in Central Japan.

**Reply to the minor revisions**

Thank you very much for providing valuable important comments. We are thankful for the time and efforts you expended. Our responses to the minor revisions are as follows:

1. *Thank you for the revision. I'd like to please ask you to consider R2's comments more fully before publication. In particular, R2 asks for an indication of why you see no correlation between d180 and temperature. I see a response in your revision notes, but not in the manuscript. This is an important point to consider as it would be reasonable, in some geographies, to expect a very strong correlation between tree ring d180 and temperature. Please re-consider that point and include a discussion of why you don't see that correlation, in the paper.*

Reply:

According to the suggestions, we add the explanation of intra-ring $\delta^{18}O$ variation for years of anomalous temperature to the end of line 153 as follows.

End of section 3.3(Lines 186-198 in the re-revised manuscript):

Other possible climatic factors affecting tree-ring $\delta^{18}O$ include spells of abnormal high/low temperature in the growing season. In the Town/City history documents used in this study, significant cold summer years recorded in $\geq 3$ titles in the 1701–1900 period are only two (1836 and 1854), and common anomalous features in intra-ring $\delta^{18}O$ variations in these two years were not found. In Central Japan, low temperature in summer season is usually accompanied by large precipitation (in this case, cold summers in 1836 and 1854 coincided with long rain and flood, respectively, according to the Town/City histories). Therefore, we decided that it was more reasonable to reflect cold summers in our research using long-rain and flood records that are more common than cold-summer records in the Town/City history documents. Records on abnormal high temperature in spring/summer/autumn are even fewer than cold summer in documentary records used in this study. Also, when we conducted correlation analysis between tree-ring $\delta^{18}O$ of our sample tree and 10-day average temperature at Gifu station, weaker correlation was found than with relative humidity or precipitation in the growing season of the sample tree. In the central part of Japan, $\delta^{18}O$ in precipitation is strongly influenced by altitude (altitude effect) and rainfall intensity (rainfall effect), and the temperature effect is marginal (e.g., Yabusaki & Tase, 2005). Therefore, it is generally difficult to read the signal of temperature variation in tree-ring $\delta^{18}O$ in Central Japan.

2. *Similarly, can I ask you to please discuss the insect damage point R2 makes, in the main manuscript, not just in your response to the reviewers.*

Reply:

According to the suggestions, we add the explanation of intra-ring $\delta^{18}O$ variation for years of insect damage to the end of line 153 as follows.

End of line 153 in the original manuscript (Lines 178-185 in the re-revised manuscript):

    Insect damage also possibly affect intra-ring $\delta^{18}O$ variation. However, there are only two years (1825 and 1836) when the insect damage was described in $\geq 3$ titles of the Town/City histories in the 1701–1900 period (Table S2). It is an insufficient number of sample years to examine the relationship with tree-ring $\delta^{18}O$. Also, we could not find common anomalous features in intra-ring $\delta^{18}O$ variations in these two years by visual comparison of the graphs. Although it is known that large outbreaks of caterpillar or sawfly affect the tree-ring width and $\delta^{18}O$ (e.g., Huang et al., 2008; Yuri et al., 2014), most of all insect damage recorded in historical documents in Japan occurred by plant hoppers or grasshoppers. Although these insects damage seriously on agricultural crops such as rice plants, they do not eat cedar leaves. Therefore, it is unlikely they affected the growth of the sample tree.

| Comment No. | Original manuscript | Revised manuscript | Re-revised manuscript |
|---|---|---|---|
| RC1-1 | Lines 118-122 | Lines 126-131
Fig. S2 | |
| RC1-2 | Lines 148-150 | Lines 161-164 | |
| RC1-3 | Lines 204-205 | Lines 271-274 | |
| RC1-4 | Lines 88-90 | Lines 92 | |
| RC1-5 | Section 3.5
(Lines 210-223
move to 3.4.4) | Lines 256-274 | |
| RC1-6 | Lines 159-160
Figure 6 caption | Lines 203-204
Figure 6 caption | |
| RC1-7 | Fig. S2 caption | Fig. S3 caption | |
| RC2-1 | | Lines 186-197 | Lines 186-198 |
| RC2-2 | | Lines 178-185 | Lines 178-185 |
| RC2-3 | | Reply to RC2 | |
| RC3-1 | Lines 106
Lines 107
Lines 151-153 | Lines 109-114
Lines 115
Lines 170-177
Fig. S1 | |
| RC3-2 | 3.4.3
Lines 145-146 | 3.4.3
Lines 158-159
Table S3 | |
| RC3-3 | Lines 88-89
Lines 129-131 | Lines 89-91
Lines 138-144 | |
| RC3-4 | Lines 265 | Lines 327-330 | |
| RC3-5 | Lines 262-263 | Lines 317-324 | |